# Economic analysis of avocado production systems: Market failures and policy distortions

Ana Luisa Velázquez-Torres[1], Francisco Ernesto Martínez-Castañeda[2],
Nathaniel Alec Rogers-Montoya[3‡], Nicolás Callejas-Juárez[4‡], Elein Hernandez[5‡*]

1 Facultad de Ciencias Agropecuarias, Universidad Autónoma del Estado de Morelos, Tecnológico Nacional de México/ TESTianguistenco, Santiago Tilapa, Mexico, 2 Instituto de Ciencias Agropecuarias y Rurales, Universidad Autónoma del Estado de México, Toluca, México, 3 Colegio de Postgraduados, Postgrado en Ganadería, Texcoco, México, 4 Facultad de Zootecnia y Ecología, Universidad Autónoma de Chihuahua, Chihuahua, México, 5 Facultad de Estudios Superiores Cuautitlán, Universidad Nacional Autónoma de México, Cuautitlán Izcalli, Estado de México, México

☙ These authors contributed equally to this work.
‡ AR-M, NC-J and EH also contributed equally to this work.
* elein_ht@comunidad.unam.mx

## Abstract

This study aimed to evaluate competitiveness and profitability at private and social prices, and assess the impact of policies and market failures on avocado production in the State of Mexico. Data were collected from 11 high-density plantations with mature trees between January and July 2021. Three groups were formed, and the Policy Analysis Matrix revealed that all Avocado Production Units (APUs) were profitable at private prices. APU 3 remained competitive when land costs were included, but none of the APUs were profitable at social prices. Indirect subsidies stabilized avocado prices, with producer subsidy rates of 3.09, 2.94, and 3.42 for APUs 1, 2, and 3, respectively. Avocado production showed a comparative advantage due to production-factor failures. As a protected product on the public agenda, avocado has high production costs.

## Introduction

Commercial openness in Mexico began in the 1980s and culminated with the signing of the North American Free Trade Agreement (NAFTA) in 1994, later renegotiated and renamed T-MEC (i.e., Mexico-United States-Canada Treaty) in 2020. T-MEC became the key agricultural policy tool, reallocating resources to boost productivity in areas where Mexico has shown international competitiveness [1]. The agreement is set for renegotiation in 2026.

Mexico leads in avocado production and exports, contributing 39% of global exports and supplying 76% of importing countries [2]. Most exports were fresh avocados (83.5%), followed by guacamole (8.9%), oil (5.5%), and pulp (2.1%) [3]. The United States represented 36% of global avocado imports [2], and 85% of Mexico's

**Data availability statement:** All relevant data are within the paper and its Supporting information files.

**Funding:** The author(s) received no specific funding for this work.

**Competing interests:** The authors have declared that no competing interests exist.

exports [3]. It ranked as the second-largest consumer worldwide, with per capita consumption of 3.3 kg, compared to Mexico with 12.3 kg [3,4].

The avocado cultivation area in Mexico showed an Average Annual Growth Rate (AAGR) of 7.7% in production from 2013 to 2023 [3]. In 2023, production volume reached 2.93 million tons, with Michoacán contributing 76%, Jalisco 11%, and the State of Mexico 4% [5]. In the State of Mexico, 12, 936 hectares were cultivated, with a production value of $2.44 million pesos. The Coatepec Harinas Rural Development District accounted for 44% of the state's production. The municipalities with the largest cultivated areas in the southern State of Mexico were Coatepec Harinas (17%), Tenancingo (6.5%), Malinalco (5.6%), Almoloya de Alquisiras (4.9%), Sultepec (2.6%), and Ocuilan with 2.3% [5]. In these municipalities, 92% of the avocado cultivated area corresponded to the Hass variety [6].

In recent years, national and international demand, along with territorial specialization, has driven the expansion of avocado cultivation into new areas, replacing forest land and maize fields [4]. This shift has brought significant changes to regional agricultural practices [6]. Avocado production is projected to increase from 2.61 to 3.16 million tons between 2024 and 2030 [7]. Avocado cultivation in Mexico produces several nationwide positive externalities, including increased domestic and export demand driven by the fruit's functional qualities, an expanded cultivated area, and substantial employment and economic gains in producing regions. Conversely, it also generates significant negative externalities: inadequate social-security coverage for farmworkers, the employment of minors and pregnant women, conversion of temperate forests and land formerly used for staple or forage crops, and price increases that have sparked territorial conflicts among criminal groups [4,6]. Paradoxically, public policies continue to prioritize and promote avocado farming as a poverty reduction strategy [8].

In Mexico, pest control campaign for avocados initiated prior to trade liberalization, aimed to support domestic and international markets. States with favorable agroecological conditions for avocado production such as: Chiapas, Colima, Guanajuato, Guerrero, Jalisco, Michoacán, Morelos, Nayarit, Oaxaca, Puebla, Querétaro, Nuevo León, the State of Mexico, Hidalgo, and Veracruz, are currently part of the program [9]. Phytosanitary programs are concentrated in the state that produces and exports the most to the USA.

The State of Mexico's government has implemented a regional productive vocation policy [10], in line with market and national strategies [11], promoting avocado production through targeted support to production units in areas with comparative advantages [12]. Since trade liberalization in 1994, when the state had 2,110 hectares, the cultivated area has grown steadily at an AAGR of 6.1%, representing an 84% increase. Currently, 88% of this area is in production, suggesting the presence of young commercial plantations [3].

Comparative advantage, a core concept in international trade theory, suggests that countries or regions should specialize in producing and exporting goods and services they can produce more efficiently, leveraging production factors such as land, labor, and capital. Conversely, they should import goods and services where they are less

efficient. Studies such as [13] evaluated the international competitiveness of avocado production using the Vollrath-Lafay methodology, determining that Mexico is a net exporter of this fruit, with no imports. Competitiveness in the European market has also been analyzed, highlighting Mexico's strong comparative advantage [14]. This advantage was furthered confirmed through international trade competitiveness indicators [15], demonstrating that Mexico's avocado production is internationally competitive. While these methodologies assess avocado performance in the global market, they do not consider the opportunity costs of agricultural production systems or the efficiency of internal resource allocation.

The Policy Analysis Matrix (PAM) methodology, within a market economy framework, aims to identify and quantify macroeconomic and sectoral policy instruments affecting agricultural competitiveness, as well as their market impacts [16,17].

This methodology has been widely applied in Mexican agriculture since the 1990s to assess trade liberalization and exchange rate policies across various crops [18,19]. Additionally, production costs, profitability, and competitiveness of avocado farming in Michoacán has been studied [20], while [21] analyzed profitability and competitiveness at private prices in the State of Mexico. Research has expanded to include policy efficiency for guava, cranberry, and berry production systems [22–24]. Internationally, [25] examined government policies for wheat production in China, and [26] studied the comparative advantage of small-scale pineapple farmers in Malaysia.

This study explores two key questions: Does the avocado-producing region in the State of Mexico possess a comparative advantage? Are public policies targeting emerging production systems with land-use changes effective in reducing poverty? To our knowledge, this is the first study to apply the full Policy Analysis Matrix (PAM) to agricultural economic policy analysis in Mexico. The study aimed to assess profitability and competitiveness at private and social prices and to examine the effects of policy and market failures on avocado (Persea americana Mill) cultivation in the Balsas agroecological transition region of Mexico.

## Materials and methods

Between January 1 to July 1, 2021, data were collected to assess the profitability and competitiveness of avocado production in the central and southeastern regions of Ocuilan (18° 52'–19° 06' N, 99° 18'–99° 29' W), where elevations range from 1,500–2,300 meters above sea level. The area has a temperature range of 8–22°C and an annual precipitation of 1,100−2,000 mm [27]. Ocuilan contributes 2.3% to the avocado belt within a dedicated agroecological zone.

### Ethics statement

The Ethics Committee of the Universidad Autónoma del Estado de Morelos approved the research protocol and data collection tools. The members of the Avocado Growers' Association were informed about the study's objectives and expected outcomes before participating. Verbal consent was obtained from the producers, with a third party present as a witness to their agreement to participate. No animals were involved in the data collection process.

### Sample size

Data were collected from 11 Avocado Production Units (APUs) out of 22 partners and 19 members of the local 'Eugenio Núñez Zetina Avocado Growers' Association' through surveys, producer interviews, and periodic visits. All APUs were small-scale plantations (< 5 ha) with low levels of technology. The sample included high-density plantations with mature, avocado-bearing trees (120–366 trees per hectare). The support policy for Mexico's avocado production system is uniform across all producers, regardless of their technology level, production volume, or climatic region.

The data collection instrument was designed around productive components, inputs, labor, fixed assets, production volumes, and prices, following the guidelines of the USDA Economic Research Service [28]. The characterization of the production system, technical parameters, and commercialization types was validated in two stages with producer participation and consensus panels. The first stage involved the full data collection process, and the second phase focused on validation with the producers' association.

## Modelling of the production systems

The 11 APUs were at a similar productive stage and employed similar technologies [17], such as interplant distances and plantation ages. They were divided into three groups based on these characteristics. Variables were analyzed at the tree level within each group, and a standardized production system was developed. Average weighted values for each APU were calculated (Equation 1) to determine the measurement unit.

Three APU's were divided into two groups, depicting a production system with the following characteristics: Group I had an interplant distance of 6 meters with age ranges between 4 and 6 years (APU 1), and between 8 and 9 years (APU 2); Group II consisted of plantations with an interplant distance of 8 meters, with an age range of between 10 and 15 years (APU 3). The production for both groups were destined to the national market. The productive area measured between 0.5 ha and 5.0 ha, with predominantly social land (not private) constituting 68% of the surface.

Commercial-scale plantations used a triangular planting setting (tres bolillo), on slopes and a rectangular grid on flat areas, with densities of 115, 160, and 366 plants/ha, respectively. These plantations were rainfed.

The commonest pests in the APU's are mites–i.e., insects that bore into tree trunks and/or branches–and thrips, while the predominant sickness is anthracnosis. The plantation is managed organically using products prepared by the producers and agrochemicals whose use is allowed in organic farming. The irrigation system is inefficient, being limited to rainwater catchment pots cladded with vulcanized mesh.

Technical-productive indicators, including production factors, marketable inputs, indirectly marketable inputs, and internal factors, were calculated at the tree level. Trees were grouped by similar characteristics, such as variety, age, and interplant distance, though superficie/area sizes ranged from 0.5 to 5.0 hectares. A weighted average, based on the number of trees per area, was applied, with the agricultural area defined as surface.

$$\frac{C_i X_i}{\sum X_i} = \frac{(Y_i \, X_i)}{\sum X_i} = (Y_i \, X_i) \frac{X_i}{\sum X_i}$$

(1)Where $C_i = Y_i X_i$: dose per surface, $Y_i$: dose per tree, and $X_i$: number of trees per lot.

Table 1 shows the PAM structure.

Private prices are ones that are received and paid by the producer in line with policy, while shadow or social prices are determined without considering public policies governing taxes, wages and prices [29]. The private budget consists of a matrix of technical coefficients pertaining to the production process, purchasing prices for production resources, and sales prices for acquired products [17]. Equation 1 shows the said prices in mathematical terms:

$$\text{Net profit} = \sum PA \, X \, A - \left( \sum PB \, YB + \sum PC \, YC \right)$$

(1)

Where A is product, B marketable inputs, C production factors, PA price of product on the regional market, PB price of the inputs used per lot in the region, Y the amount of PAM inputs, and X yield in tons per lot.

**Table 1. Policy Analysis Matrix Structure according to competitiveness and profitability performance.**

| Concept | Income | Production costs | | Profit |
| --- | --- | --- | --- | --- |
| | | Commercial inputs | | Internal factors |
| Private prices | A | B | C | D |
| Social prices | E | F | G | H |
| Policy effect | I | J | K | L |

Source: [17].

Marketable inputs are defined as inputs such as fertilizers, insecticides, herbicides, seeds, etc. that can be acquired on both the national and the international markets, while indirectly marketable inputs are defined as inputs with both a marketable component and an internal factor component such as agricultural machinery and pumping equipment. Finally, internal factors such as labor, land, water and electric power are ones that are not quoted on the international market because, while they cannot be physically exchanged between countries, they constitute inputs in the production process.

Social prices were calculated using the methodology proposed by [30]. The cost of capital was calculated based on the working-capital-yield rate (marketable inputs + labor). The development-bank rate of 13.5% for 2020 was considered, equivalent to the Interbank Equilibrium Interest Rate (Spanish acronym: TIIE) of 4.5%, with a historical inflation for the period of 4.1 + 5 prime points. The rate used for shadow prices was 3.2% [28]. Finally, the land tax of USD $85.66 per surface was obtained based on the cadastral value of the studied zone.

The production costs include a 9% tax on toxic pesticides, the Special Tax on Production and Services (Spanish acronym: IEPS) for both production and commercialization –5.3% on gasoline and 5.4% on diesel, plus 16% sales tax– for 2020. The values of direct subsidies at shadow prices considered a 16% reimbursement of the sales tax stemming from the purchase of inputs to produce agrochemicals, packaging, fuels and lubricants. The divergences were obtained via the difference between private and social prices, and their size shows the extent to which distorted private prices differ from social or efficiency prices. We referred to the respective worldwide prices and to the importation ones to calculate the efficiency prices of marketable products and inputs [31], the equivalents of which assume that international market prices reflect the opportunity costs of the production and value arising from shortages at the consumer level, being close to the costs that would prevail if competition existed [32].

In order to come up with an equivalent social price, the prices of imported inputs were calculated based on prices at the Mexico-US border or CIF (Cost Insurance Freight), as well as the prices for transportation within Mexico, it being deemed that the point of entry was the Port of Veracruz, and that production took place in the central agricultural region of Toluca in the State of Mexico. Both analyzed groups focused on the domestic market, as the product lacks certification for sanitary handling and transport [9]. The prices at the Mexican border in foreign currency were converted into social prices, in Mexican pesos, by correcting or adjusting the exchange rate in line with the pertinent degree of overvaluation or undervaluation and eliminating the pertinent customs duties at the border, along with internal subsidies or taxes.

The product's export price was based on the FOB (i.e., Free on Board) price, which was similar to the CIF price. The product was exported to the USA via New Mexico and the cost of the internal or primary production factors was calculated based on the internal opportunity costs. The equivalent of the social price of equipment and machinery was calculated based on the capital recovery factor, adjusted based on the equilibrium interest rate of the economy. It was assumed, in the case of manual and mechanized labor, that the social price was the same as the private prices paid on the local market. The protection, efficiency, transfer, profitability, and sales-tax rates were derived from the PAM.

The currency values were converted into US Dollars at the exchange rate of USD 1: MXP 19.8455 published by the Bank of Mexico at the close of business on July 30th, 2021.

## Private profitability

Private profitability, defined as $D = A-(B + C)$, it is a measure of competitiveness with current private prices. $E = $ Social profitability, is defined as $H = E-(F + G)$, a measure of efficiency or comparative in price efficiency. Transfer due to product price ($I = A-E$) is either a potentially positive divergence caused by an implicit subsidy or a transfer of resources to the agricultural system, or a potentially negative divergence that leads to an implicit tax or a transfer of resources outside the system. Transfers by inputs ($J = B-F$). The divergence between marketable inputs generated by private costs, B, minus social costs, F, leads to a transfer of marketable inputs. The latter divergence may either be positive (generating an implicit tax or transfer of resources outside the system) or negative (generating an implicit subsidy or transfer of resources to the agriculture system). The divergences may influence the prices of the production factors (qualified labor, unqualified labor,

capital and land). The divergence of market factors causes private-factor costs (C) to differ from social costs. The difference between private-factors (C) and internal-factor costs leads to a transfer of factors (K = (C – G)). The said divergence may be either positive (generating an implicit tax or a transfer of resources outside the system) or negative (generating an implicit subsidy or transfer of resources to the agriculture system). Total transfers (L = I-J-K); Total effect of policies (L = D-H).

## Coefficients for policy analysis

The private profitability coefficient (PPC) indicates the amount of extraordinary profit obtained by the producer as a function of total cost [20] and measures the system's competitiveness and profitability at private prices, being expressed as a coefficient of total profit divided by the sum of the total costs. CRP = D/ B + C.

The Nominal-Protection Coefficient on Output (NPCO), which refers to the level of protection of the main product, measures the impact at market (gross) prices of government protection of agricultural products.

If the NPCO is > 1, the system takes advantage of the protection, if the NPCO is < 1, the system is subject to taxation, and if the NPCO = 1, income at private prices is the same as income at social prices, meaning there are no policy effects or NPCO, where NPCO is the relationship between income at private prices (A) and income at social prices (E). NPCO = A/E.

The Nominal-Protection Coefficient for Marketable Inputs, NPCI, which is free of foreign-exchange or basic-product distinctions, is used to compare marketable inputs (e.g., fertilizers and fuel). The ratio formulated in order to measure marketable-input transfers is expressed as NPCI = B/F, showing the cost of marketable inputs and the difference between private and social prices. If the NPCI is > 1, the cost of domestic inputs is higher than the cost of inputs at worldwide prices, and the system is taxed/aggravated in accordance with the policy. If the NPCI is < 1, the market price of the inputs is lower than the prices that would prevail in the absence of policies. Moreover, it reveals the presence of a subsidy or tax, along with restrictions on trade that increase or reduce prices or an overvalued or undervalued exchange rate.

The effective protection coefficient, EPC, show the general level of protection, taking account of the impact of policies on the value of the products and marketable inputs. This coefficient shows the joint effects of policy transfers that affect both marketable products and marketable inputs. The ratio expressed as EPC = A – B/ E – F is the ratio between added value at private-market prices (A – B) and added value at social market costs (E – F) [16,33]. The EPC, which is an incentive index that is determined based on the ratio between added value at market prices and social or efficiency prices [17], includes both transfers to the product and marketable inputs [34]. If the result is < 1, the effective prices faced by producers do not reflect support transfers by market and exchange-rate policies applicable to both the product and the inputs – i.e., there is no transfer of the policies applicable to both the product and the marketable inputs [32].

The production-factor coefficient, PFC, which measures the efficiency or comparative advantage of crop production, shows whether the country has a comparative advantage, indicating that the value of the production factors used to grow the crop is higher than their added value. The PFC will always be positive unless the added social value of growing a crop is negative. If it is < 1, the system has a comparative advantage, indicating the use of local resources that are cheaper than global ones. If the PFC is > 1, the system has no comparative advantage and social profitability is negative.

The Producers' Subsidy Rate (PSR) is the index reflecting policies/change of market distortions of the system's total income at social prices – the size of the difference between the reference system at social prices and the current system at local-market prices. The purpose of the said index is to show the level of transfers from divergences as a proportion of the undistorted value of the system's income [17]. The PSR, which shows the extent to which a system's income has increased or diminished due to policy, is the result of dividing net policy transfer (L) by income at social prices (E).

The equivalent of the product subsidy (PSE) that is defined as the index of policy reflection/market distortions for increasing or reducing the system's total income at local-market prices. When the PSE is positive, this indicates that the

policy subsidizes producers, and when it is negative, this indicates that the policy supports consumers. It was calculated dividing (L) by income at private prices (A) [35].

The profitability coefficient (PC) is a measure of the extent to which policy affects the system's profitability. If the PC is > 1, the system benefits from the sector's net transfers. If PC is < 1, the system benefits from the system's net transfers, where the ratio of profits at private prices (D) is compared to the benefit of social prices (H).

## Results

### Yield

Plantations aged 4–6 years yielded the lowest average (62 kg/tree), consistent with values reported for young plantations in Michoacán, which have not yet reached their peak productivity [20]. Meanwhile, 8-year-old plantations with a 6-meter spacing yielded 144 kg/tree, comparable to previous findings from the State of Mexico [21]. Lastly, 10-year-old trees yielded 115 kg/tree [20], similar to commercial plantations reported in Jalisco and Michoacán [20,36].

Table 2 shows weighted standard deviation based on technical coefficients of avocado production in Ocuilan.

Product quality was classified following Mexican standard NMX-FF-016-SCFI-2016 [37], which establishes fruit categories based on size or weight (Table 3). Fruits harvested from developing trees (4–6 years old) consisted of 20% Extra-class or first-class, 40% Class-1 (second-class), 20% medium-class, and the remainder classified as commercial and marble-class. For plantations aged 8–15 years, regardless of planting density, fruit quality distribution was 60% Extra-class, 20% Class-1, and the remainder classified as second-class or commercial. The product price ranged from USD

Table 2. Weighted standard deviation based on technical coefficients based on technical coefficients of avocado production.

| Plantation | 6*6 m | | | | 8*8 m | |
|---|---|---|---|---|---|---|
| Age of avocado trees (years) | 4–6 | | 8–9 | | 10–15 | |
| | $\bar{x}_w$ | $S_w$ | $\bar{x}_w$ | $S_w$ | $\bar{x}_w$ | $S_w$ |
| B. Marketable Inputs | 5,643.24 | 1,488.42 | 6,820.81 | 2,649.76 | 6,899.50 | 3,556.38 |
| Fertilizers (kg/ha) | 5,262.24 | 1,369.08 | 6,464.62 | 2,439.77 | 6,535.15 | 3,290.06 |
| Fungicides (kg/ha) | 38.59 | 12.85 | 13.73 | 13.73 | 35.71 | 36.45 |
| Insecticides (kg/ha) | 7.88 | 6.79 | 23.34 | 13.40 | 11.47 | 13.69 |
| Acaricide (kg/ha) | 52.57 | 24.45 | 37.52 | 30.60 | 126.40 | 134.27 |
| Gasoline (L/ha) | 167.36 | 44.96 | 200.30 | 82.59 | 165.01 | 71.22 |
| Lubricants (L/ha) | 25.10 | 6.74 | 30.05 | 12.39 | 24.75 | 10.68 |
| Packaging (sacks/ha) | 88.51 | 23.54 | 50.25 | 57.28 | 0.00 | 0.00 |
| Others | 1.00 | 0.00 | 1.00 | 0.00 | 1.00 | 0.00 |
| C1. Production Factors | 45.90 | 4.18 | 45.01 | 1.99 | 43.69 | 4.49 |
| Direct labor (workday/ha) | 9.88 | 1.35 | 9.00 | 0.00 | 9.00 | 0.00 |
| Mechanized labor (machine workday/ha) | 17.00 | 0.00 | 17.00 | 0.00 | 17.00 | 0.00 |
| Operating loan (USD $/ha) | 0.01 | 0.94 | 0.00 | 0.00 | 0.00 | 1.00 |
| Land (ha) | 0.01 | 0.94 | 0.01 | 0.99 | 1.00 | 0.00 |
| Transportation (USD $/ha) | 7.00 | 0.00 | 7.00 | 0.00 | 3.69 | 3.49 |
| Water fee (USD $/ha) | 0.01 | 0.94 | 0.00 | 1.00 | 1.00 | 0.00 |
| Technical assistance (USD $/ha) | 12.00 | 0.00 | 12.00 | 0.00 | 12.00 | 0.00 |
| C2. Indirectly marketable inputs | 167.56 | 45.04 | 179.94 | 73.99 | 52.88 | 0.00 |
| C. Total of C1 and C2 | 213.46 | 49.22 | 224.95 | 75.98 | 96.57 | 4.49 |
| Total income (USD $/ha) | 11,063.35 | 2,942.91 | 15,236.84 | 5,552.07 | 15,399.05 | 8,347.01 |

$\bar{x}_w$ = Weighted mean; $S_w$ = Weighted standard deviation.

**Table 3. Fruit categories according to Mexican classification of avocado.**

| Caliber | Weight (g) |
|---|---|
| **Super** | > 266 |
| **Extra** | 211-265 |
| **Class 1** | 171-210 |
| **Medium** | 136-170 |
| **Commercial** | 85-135 |
| **Marble** | < 85 |

Source: [37].

$0.66 to $1.01/kg, depending on fruit size categories defined as follows: Super (>265 g), Extra-class (211–265 g), Class-1 (171–210 g), Medium-class (136–170 g), Commercial-class (85–135 g), and Marble-class (<85 g).

**Analysis of profitability, competitiveness, policy effects, and market failure.** APU 1 had a positive income. However, it was not profitable at private and social prices, as a result, it showed the highest profit divergences in comparison with APU 2 and APU 3.

Similarly to APU1, APU 2 had a positive income, being profitable at private prices but not at social prices, and with less profit divergences.

Even though APU 3 had the highest income, it was still only profitable at private prices. APU 3 showed the smallest profit divergences in comparison with APU 1 and APU 2 (Fig 1).

## Profitability in avocado cultivation

Though APU's 1 and 2 were profitable at private prices (D) excluding land costs, with remuneration to the system being USD$1,134.51 and USD$2,186.69 respectively, when land costs were included, they were not profitable. APU 3 was profitable at private prices regardless of whether land costs were included or not, yielding USD$3,379.26 without considering land costs, and USD$1,867.59 (Table 4).

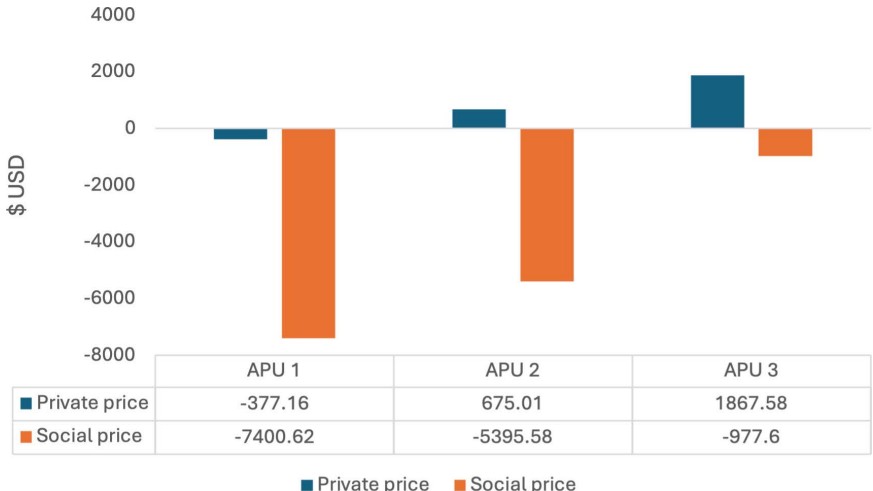

| | APU 1 | APU 2 | APU 3 |
|---|---|---|---|
| **Private price** | -377.16 | 675.01 | 1867.58 |
| **Social price** | -7400.62 | -5395.58 | -977.6 |

■ Private price  ■ Social price

**Fig 1. Profitability by tree age in three Avocado Production Units (APUs) in $USD.** Source: author produced, using field data.

**Table 4. Profit in USD generated by three Avocado Production Units (APUs).**

| Concept | Interplant distance (m) | Age of the avocado trees (years) | Profit excluding land cost | | Profit including land cost | |
|---|---|---|---|---|---|---|
| | | | PP ($) | SP ($) | PP ($) | SP ($) |
| APU 1 | 6*6 | 4–6 | 1,134.51 | −2,678.80 | −377.16 | −4,1048.81 |
| APU 2 | 6*6 | 8–9 | 2,186.69 | −3798.22 | −675.01 | −5,2242.24 |
| APU 3 | 8*8 | 10–15 | 3,379.26 | −1,484.84 | 1,867.59 | −2,910.85 |

Private Price (PP), Social Price (SP). Source: author produced, using field data.

## Distorting policies and market failures

The greatest variation was in income (I). The effective prices differ from the efficiency ones in APU 1, APU 2 and APU 3 by 71%, 73% and 93% respectively. An average price of USD$0.76/kg was paid for the on-farm product on the internal market during 2020, while the Free-on-Board price (FOB), without considering policy effects (i.e., reimbursement of Sales Tax) was USD$0.20/kg.

## Transfer of marketable inputs

The respective costs of marketable inputs for APUs 2, 3 and 4 were USD$728.63, USD$1,431.53 and USD$5,508.25. The main outlays, constituting 43%, 48%, 55% of total costs respectively, were for fertilizers (Fig 2).

## Transfer of production factors

The variation generated by production factors (K) in the APUs was due to the costs of land and capital. Though the cost of land for the three production systems constituted 48% of the production-factor resources, there is no other cultivation

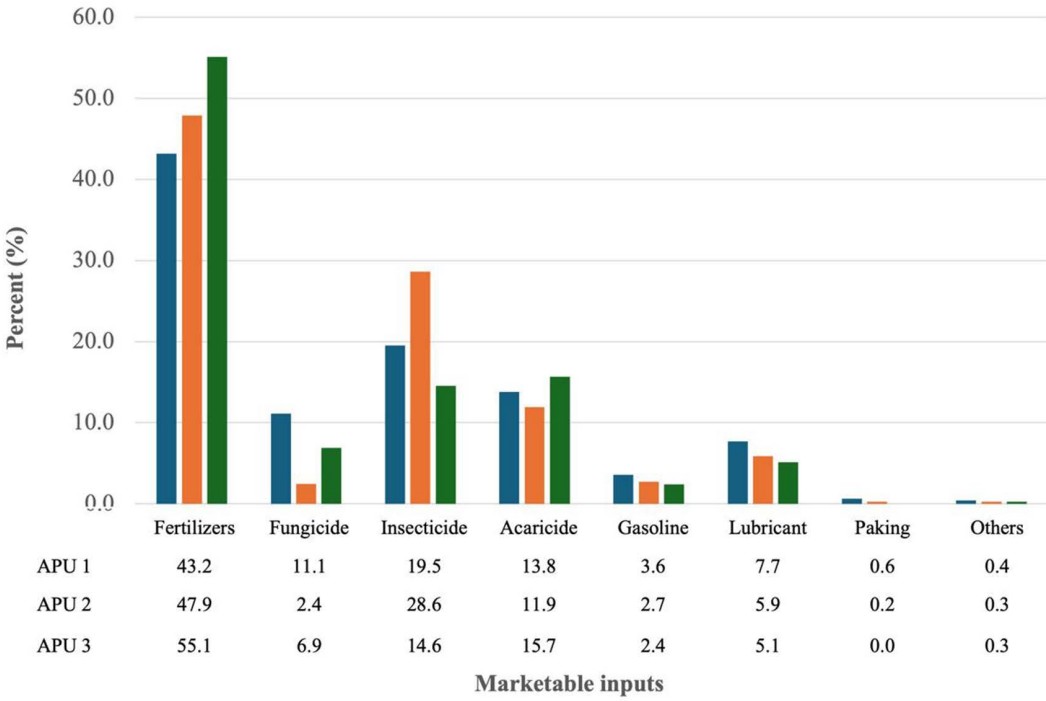

**Fig 2. Marketable inputs in three Avocado Production Units (APUs). Source: author produced, using field data.**

available that makes it possible to determine the opportunity cost of land use. For its part, the social opportunity cost of capital constituted 3.2%, being lower than the 13% annual private interest rate for working capital.

## The effects of policy on the avocado-production systems

The results of the PAM made it possible to carry out an analysis for the purpose of planning product prices, public investment, and agricultural-research policies. The following coefficients are used to compare policy variations or market failures and their transfers to agricultural products in absolute terms (Table 5).

### The private-profitability coefficient (PPC)

While APU 1 was neither profitable nor competitive at social prices, APU 2 (6%) was competitive but not profitable compared to the opportunity cost evaluated at an annual rate of 13.5%. For its part, APU 3 was both competitive and profitable at private prices, reflecting a capital opportunity cost 4.5% higher than the interest rate.

### Coefficient of nominal product protection (NPCO)

This coefficient was more than one, showing that market prices are higher than social ones (Table 5). This variable confirmed that APU's 1, 2 and 3 received subsidies or transfers to the product price of 243%, 272% and 1,391% respectively during 2020.

### Nominal-protection coefficient for marketable inputs (NPCI)

This indicator was higher than one. The prices of marketable inputs were higher than the cost of inputs at worldwide prices and the system is aggravated by the policy. This coefficient revealed that production was implicitly taxed on agrochemicals and fertilizers in the respective amounts of 17%, 25% and 310% for APU's 1, 2 and 3. This increased the production cost as against the social prices. This expenditure pertained to customs-tariff transfers due to agrochemical toxicity, the Special Tax on Products and Services, and Sales Tax.

### Effective protection coefficient (EPC)

In absolute terms, APU's 1 and 2 had respective transfers of 53% and 30% stemming from the set of current policies (Table 5). For its part, APU 3 had a 442% transfer that can be attributed to the fruit in question which was of export quality (>265 grs.)

**Table 5. Economic competitiveness and efficiency of three Avocado Production Units (APUs).**

| Coefficient/ | Group I | | Group II |
|---|---|---|---|
| | APU 1 | APU 2 | APU 3 |
| Interplant distance | 6 m | | 8 m |
| Age of avocado trees | 4–6 | 8–9 | 10–15 |
| PPC | −0.05 | 0.06 | 0.18 |
| NPCO | 3.43 | 3.72 | 14.91 |
| NPCI | 1.17 | 1.25 | 4.10 |
| EPC | −1.53 | −1.30 | −5.42 |
| PFC | 0.0 | −0.24 | −0.54 |
| PSR | 3.09 | 2.94 | 3.42 |
| PSE | 0.90 | 0.79 | 0.23 |
| PC | 0.0 | −0.13 | −0.90 |

Source: author produced, using field data.

### Production-factor coefficient (PFC)

Since the PFC was less than one, showing that avocado production enjoyed a comparative advantage. The cost of domestic resources (CDR) indicated that APU's 2 and 3 had respective comparative advantages of −0.24 and −0.54 that stemmed from the payment of wages without social benefits (Table 5).

### Policy-reflection/market-distortion index (PSR)

This result shows the policy originated divergences that have generated respective net transfers of 309%, 294% and 342% to APU's 1, 2 and 3. If all the policies pertaining to inputs and factors were eliminated, the avocado system's NPCO would have to be increased to 342% in APU 3 in order to enable the system to keep the same level of private benefits.

### Policy-reflection/market-distortion index (PSE)

This coefficient showed that the Government should provide subsidies to the producers in APU's 1, 2 and 3 amounting to 90%, 79% and 23% of their respective internal costs (Table 5) in order to keep them profitable.

### Profitability coefficient (PC)

This coefficient measured the impact of all the transfers on private benefits. It constitutes an expansion of the EPC to include factor costs (along with income, marketable securities and input costs). The evaluation results showed that there is no distortion in the product prices in APU 1 because any possible distortion was offset by equal and opposite input-price distortions; APU 2 lost 13% of its profits, while APU 3 lost all its income plus 90% of its profits due to the policy that was in force when the study was carried out (Table 5).

## Discussion

### Modelled avocado-production units (APUs)

According to [38] and [20], the plantations are classified as a small-scale avocado production units with low technification. Avocado production on a commercial scale is relatively new in the municipality of Ocuilan, and advanced plantations more than 30 years old pertain to back-yard lots with genetic varieties such as Hass, Hass-Fuerte, and, to a small extent, interspersed with creole trees. A clear trend has emerged favoring the establishment of new commercial Hass plantations, often at the expense of native varieties, as noted by [39]. This information accords with that published by [40], for commercial plantations in the south of the State of Mexico. For their part, commercial-scale plantations are developing ones (>4 years old) and young ones (>20 years old).

The most populated surface follows a regional pattern observed during the last decade, with Hass varieties being preferred due to the introduction of varieties such as Hass-Méndez or Méndez-Mejorado. This phenomenon has been documented on avocado plantations in the State of Mexico, the State of Morelos, and the State of Michoacán [20,38,40]. Developing avocado plantations are often interspersed with other crops or fruit trees as an income diversification strategy, given that avocado production has not yet become profitable.

The studies by [20,38,41] reported a similar scant or limited technology level in Morelos, Michoacán and Ethiopia.

The rain-fed water regime is similar to the one in production units in the State of Mexico reported by [39], with an inefficient irrigation system limited to rainwater catchment systems by traditional methods.

### Yield

The yield per hectare is 9.0 tons, lower than the average yield of 11.25 kg per tree in the State of Mexico in 2020 [3], while the national yield reached 11.70 tons per hectare. These production volumes are similar to those reported by [20] and [42].

### Analysis of profitability, competitiveness, policy effects, and market failure

The PAM showed the difference between private and social prices, with the result being the degree of efficiency of the resources that influence production and productivity, a divergence that could have been caused by the policies implemented in the sector or by market failures [17].

### Profitability in avocado cultivation

The positive levels of profitability calculated excluding land costs are due to efficiency in the use of production factors, as well as to the fact that the plants are reaching productive maturity, which implies an increase in production volumes, and also in selling prices due to the higher caliber of the fruit, mainly in APU's 2 and 3. The profitability data presented in this study are similar to the ones reported by [43] for developing production unit APU 1. The aforesaid authors carried out a financial study of the avocado-production system using production data in similar regions, reporting an estimated profitability (net present value) of USD$848.73 for 2019, with an optimistic scenario of USD$1,181.76 and a pessimistic one of USD$333.60.

One of the main implications of positive private profitability was that the plantations in question were at a development stage [17]. The low level of profitability of APU 1 is due to the production volume and caliber of the fruit, which had the lowest price of USD$0.70 per kg, due to the fact that the trees had not reached their full productive level. However, none of the APU's was profitable at social prices with the policy currently in force, regardless of whether land costs were included or not.

### Distorting policies and market failures

The results obtained in this study reflect protection or subsidies and can be attributed to the policies applying to export products.

In 2020, the average farm gate price in the domestic market was MX$15.00/kg, while the export parity price (FOB), after adjusting for policy effects (VAT refunds), was MX$4.00/kg.

This effect is linked to the cascading impact of indirect taxes like VAT and IEPS on production processes, which are not adequately offset by tax refund subsidies. This creates challenges in balancing efficiency and equity in transfers.

### Transfer of marketable inputs

The variation in marketable inputs (J) was due to the fiscal policies applying to transportation, fuel, sales tax and direct subsidies. As also reported by [20], this was the second biggest outlay after pesticides, whose excessive use can be attributed to ignorance about the avocado plant's nutritional needs [44].

These data are indicative of production costs that exceed those on the international market, obliging local consumers to pay more per kilo of avocado. The fact that the production cost is higher than the social one indicates that the policy is providing a positive transfer, causing the production system to make bigger profits and cover costs that are higher than the private ones that prevail when no subsidies are provided. This means that the protectionist policy currently in force leads to lower levels of profit on avocado sales.

### Transfer of production factors

The findings of this research showed a transfer outside the system of factors amounting to 61.5% of the total resources in accordance with the policy currently in force, while, since unqualified-labor costs were insignificant, with the cost of such labor amounting to 3%, qualified labor amounting to 5%, and specialized labor amounting to 4%, it is assumed that the private wage rate was an indicator of the social rate for this type of wage.

The respective net transfers (L) for APU 1, APU 2 and APU 3 of USD$7,023.46, USD$6,070.60 and USD$2.845.18 per hectare were calculated based on the transfers inside and outside the system pertaining to the refunding of sales tax on

purchases of fertilizers, agrochemicals, packing materials, gasoline and diesel fuel, and lubricants. This variable represented a policy transfer of both inputs and outputs from the system.

## The effects of policy on the avocado-production systems

The results of the PAM made it possible to carry out an analysis for the purpose of planning product prices, public investment, and agricultural-research policies. The following coefficients are used to compare policy variations or market failures and their transfers to agricultural products in absolute terms.

The obtained NPCO coefficient showed that that internal price of avocado was higher than the world-market price [35], so that the said product did not generate any profit and, indeed, within the analyzed framework, constituted an expense for the State. This type of policy focuses on generating foreign exchange to strengthen the economy and facilitate the acquisition of essential goods [16]. NPCO reflected government support for avocado production. Although growers obtain prices that are higher than the international ones with this incentive, consumers get negative protection, because they must pay a higher price due to the government policy that is in force [16,34,35]. As asserted by [45], the lack of consumer protection is due to the tax on fuel, the Special Tax on Production and Services, and the Sales Tax, since, given that the latter two are indirect taxes, the end national consumer is the one that has to pay them. These indirect taxes include the Special Tax on Production and Services (IEPS), applied since the 1980s to specific goods such as beer, tobacco, gasoline, and diesel; the Value Added Tax (IVA), applied broadly across goods and services [23]; and the toxicity tax applied to agrochemicals. Such taxes are integrated into consumer prices and do not require explicit tax reporting. Although indirect taxes affect all consumers regardless of income, their impact is greater on lower-income populations, particularly affecting small-scale producers who operate informally and thus lack access to tax reimbursement mechanisms.

According to [24], PSE reflects a big incentive to produce avocados rather than other crops, and that avocado growing is paramount for the international commerce of the State of Mexico.

Regarding PC, [20] concur that the profitability and competitiveness of APU's 1, 2 and 3 at private prices, excluding land costs, are mainly due to production-factor efficiencies, as well as the physical maturity of the trees, production volume and product caliber, all of which determined the avocados' final price. The combination of production, economic and physiological factors may not always benefit the avocado product system.

According to [12], when the TLCAN was signed, commercial customs duties were eliminated and subsidies gradually abolished. Small-scale producers bore the tax burden, incurring expenses through indirect taxes on production inputs such as fertilizers, agrochemicals, packaging, and fuel. Operating informally, they did not benefit from VAT refund incentives. The findings indicate that indirect subsidies, such as tax refunds for exporters, sustain avocado prices in the international market, driven by distortionary policies. Such subsidies are common in emerging economies to maintain international competitiveness and accelerate income growth rates [17,46].

Distorting policies affecting avocado cultivation have artificially raised domestic agricultural prices above international levels [47]. This type of support harms low-income consumers [16,17], widens the income gap between small and large-scale farms, and undermines the competitiveness of the food sector [47].

In 2019, Mexico's agricultural subsidies accounted for 65.4%, while those of its trading partners, the United States and Canada, were 7.3% higher during 2019–2020 [48]. Although the scale of transfers is comparable, trading partners allocate them to support services such as research, training, and dissemination, whereas Mexico directs its transfers toward price support, limiting productivity growth and creating resource insufficiencies in the agricultural sector [49].

Small producers are excluded from export markets due to their informal operations and limited resources. However, small-scale production is significant as it meets the domestic demand for this fruit. Emerging countries also face higher trade costs when exporting compared to developed countries [50]. However, the results of this study show that it is indirect subsidies that regulate the price of avocados on the international market. In this regard, it is common for emerging countries to subsidize market prices in order to stay competitive at the international level and speed up their income-growth

rate [17,46]. As mentioned by [47], the distortion caused by the policies governing avocado cultivation has caused the internal prices of agricultural products to artificially exceed international ones. This kind of support harms consumers [16,17] –especially those with less resources– and widens the income gap between small agro-businesses and large ones, as well as reducing food-sector competitiveness [47].

In line with the conclusions of [51–53], who report that their results show that avocado-growing in Mexico is very competitive at the international level, our results show that APU's 2 and 3 enjoyed a comparative advantage at the said level. However, in concordance with [50], such comparative advantage was due to market failures and production factors such as natural-resource availability, low wages and the absence of worker benefits. It bears pointing out that, since the study was carried out in APU's with growing trees that were still establishing themselves, the policies currently in force may not be sufficiently sensitive and wide-ranging to benefit such systems, under the said conditions. An effect of market failure is the concentration of avocado exports in Michoacán (over 70%) and Jalisco. [54] highlights that 45–80% of exports to the United States are managed by transnational companies, leading to wealth concentration and significant income inequality for local producers. [50] adds that Mexico's comparative advantage relies on factors like natural resources, low wages, and a lack of social benefits such as education and healthcare, reflecting multidimensional poverty beyond mere economic deprivation. Market failures associated with limited human capital accumulation are largely institutional, stemming from low levels of technological adoption and inefficient financial systems. Addressing these challenges requires public policies that strengthen education systems and support robust research infrastructure to drive development [54]. In this regard, targeted interventions are essential to overcome structural market failures that constrain economic growth and long-term development.

The general subsidies that were in place in Mexico in 2019 amounted to 65.4%, while those implemented in 2019 and 2020 were comparatively bigger [48]. In this regard, [49] state that, although the subsidies and transfers implemented by Mexico are almost equal to those put in place by its trading partners, the said partner's subsidies pertained to support activities such as research, training, and dissemination, while those implemented by Mexico are aimed at supporting prices rather than boosting productivity, thus resulting in a scarcity of resources in the countryside, due to defective assignation. In an export context, small producers with limited resources just cannot compete, since, unlike developed countries, they face higher costs than their competitors [50].

The methodology used to study the production system, with the determining features prevailing at the time when the study was carried out, enabled us to measure the levels of competitiveness in different APU's as well as policy variations and their effects.

The three APUs were profitable and competitive at private prices when the cost of land was not taken into account. At social prices, it was shown that avocado cultivation enjoys a comparative advantage due to market failures in production factors that did not reflect their real scarcity value (high interest rates and low wages). Market failure in human-capital accumulation stems from an intergenerational process that manifests in different life stages of agricultural laborers: early childhood (health and nutrition); youth (nutrition and parental income); and adulthood (health). Knowledge applied to economic activities is therefore a key driver of economic growth [54]. In the study area, the avocado labor market is characterized by harvest seasonality, outdated production techniques and limited value added to the final product, all of which translate into low wages. By contrast, exporting states encompass the entire value chain—from primary production to export-oriented agro-industry—and create linkages with the domestic market through better-paid jobs. Higher wages are possible because value is added along the chain and a more educated workforce is employed [53].

Since avocados, which were seen to be a highly protected product that forms part of the government agenda as net-export product, had high production costs in the evaluated systems. It was necessary to protect product prices on the international market due to distorting policies rather than market failures. Transfers of sales tax on fuel, agrochemicals, fertilizers and packing materials resulted in an implicit financial burden on the consumer due to the taxes payable for agrochemical toxicity, the Special Tax on Products and Services and the Sales Tax, increasing the internal product price. The

authors acknowledge the limitations of this study, as the sample was restricted to a single municipality in the State of Mexico due to the COVID-19 health emergency, focusing on an emerging avocado production area for domestic consumption.

## Conclusions

To assess whether the avocado-producing region of the State of Mexico has a comparative advantage and if public policies for emerging production systems with land-use changes are effective, the profitability and competitiveness of the avocado production system were evaluated, revealing market failures and distortionary policies.

The three APUs from groups I and II were profitable and competitive at private prices, excluding land costs. As the production systems have not yet reached full maturity, this indicator suggests that commercial avocado production in Ocuilan is likely to expand in the coming years.

At social prices, avocado cultivation shows a comparative advantage due to market failures, where production factors fail to reflect their true scarcity value (e.g., high interest rates and low wages). This undervaluation makes production in Ocuilan cheaper than in regions with higher production costs.

The protection of international market prices stems from indirect taxes on inputs and subsidies through tax burdens, creating a distortionary policy. For small-scale producers, who are not registered in the tax system, do not claim tax refunds, but still pay indirect taxes, this policy becomes even more inefficient, further increasing inequality. Market failures have resulted in both positive and negative externalities: on the positive side, they generate employment and support the regional economy; on the negative side, they drive land-use changes, erode food sovereignty and biodiversity, deplete natural resources, and contribute to global warming.

Avocados for the international market are highly protected and prioritized as a key export product, yet they incur significant social, economic, and environmental costs due to market failures. In contrast, avocados for the domestic market face a heavy tax burden from indirect taxes, which are passed on to local consumers.

More research is needed to assess the multidimensional poverty associated with avocado production in both export-focused and domestic consumption areas. More detailed analyses of fiscal policies for industrial export operations should also be conducted, given the limitations of this study.

Further analysis is needed to assess the feasibility of implementing specific policies for the APUs studied, to promote productivity, research, technology transfer, human capital development, and meet the 2030 environmental agenda benchmarks.

The results demonstrate how policies, like those in Mexico for these systems, can impose social costs by shifting the burden of government transfers to consumers.

## Supporting information

**S1 File. Avocado.**
(XLSX)

## Acknowledgments

The authors wish to thank the members of the Eugenio Núñez Zetina Avocado Growers' Association that kindly provided the information used in this study, and the Autonomous University of the State of Morelos.

## Author contributions

**Conceptualization:** Ana Luisa Velázquez-Torres, Francisco Ernesto Martínez-Castañeda, Nicolás Callejas-Juárez, Elein Hernandez.

**Data curation:** Ana Luisa Velázquez-Torres, Francisco Ernesto Martínez-Castañeda, Nicolás Callejas-Juárez.

**Formal analysis:** Ana Luisa Velázquez-Torres, Francisco Ernesto Martínez-Castañeda, Nicolás Callejas-Juárez.

**Investigation:** Ana Luisa Velázquez-Torres.

**Methodology:** Ana Luisa Velázquez-Torres, Nicolás Callejas-Juárez.

**Resources:** Nathaniel Alec Rogers-Montoya, Nicolás Callejas-Juárez.

**Supervision:** Francisco Ernesto Martínez-Castañeda, Nicolás Callejas-Juárez.

**Validation:** Francisco Ernesto Martínez-Castañeda, Nicolás Callejas-Juárez.

**Visualization:** Francisco Ernesto Martínez-Castañeda, Nathaniel Alec Rogers-Montoya, Elein Hernandez.

**Writing – original draft:** Ana Luisa Velázquez-Torres, Francisco Ernesto Martínez-Castañeda, Nathaniel Alec Rogers-Montoya, Elein Hernandez.

**Writing – review & editing:** Ana Luisa Velázquez-Torres, Francisco Ernesto Martínez-Castañeda, Nathaniel Alec Rogers-Montoya, Elein Hernandez.

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
