## [Decision Letter · Decision Letter 0]

8 Dec 2024

Dear Dr. Hernandez,

Thank you for submitting your manuscript to PLOS ONE. After careful consideration, we feel that it has merit but does not fully meet PLOS ONE’s publication criteria as it currently stands. Therefore, we invite you to submit a revised version of the manuscript that addresses the points raised during the review process.

**ACADEMIC EDITOR:**

Comments to the Author

1. Is the manuscript technically sound, and do the data support the conclusions? The manuscript must describe a technically sound piece of scientific research with data that supports the conclusions. Experiments must have been conducted rigorously, with appropriate controls, replication, and sample sizes. The conclusions must be drawn appropriately based on the data presented.

Reviewer #1: Partly

Reviewer #2: Yes

2. Has the statistical analysis been performed appropriately and rigorously?

Reviewer #1: Yes

Reviewer #2: Yes

3. Have the authors made all data underlying the findings in their manuscript fully available?

Reviewer #1: Yes

Reviewer #2: Yes

4. Is the manuscript presented in an intelligible fashion and written in standard English?

Reviewer #1: Yes

Reviewer #2: Yes

5. Review Comments to the Author

Reviewer #1: Abstract: ok

Introduction: Line 33 Don't you have a more up-to-date figure?

The topic should be delved into if there are studies related to the cultivation of avocado, at the Mexico and state level, or related studies to see the methodological viability in important crops, in Michoacán or Jalisco.

Material and methods: Line 52 – 54 add a location map. Delve into the place of study, if it is important in the production of avocado, what ranking it occupies in the state, among other characteristics that describe the relevance of the crop in the place.

Line 64 What was the density?

Line 115 – 116 Is the product produced in this area imported to the USA? How much?

It is understood that the avocado produced at this site is the same as that produced in Michoacán. The characteristics of avocados in both areas must be different. These differences must be clarified.

On the political side, it is homogenized in the country.

Results: line 201 – 216 this part should go in methodology.

Line 218-220 compare it with the production of Michoacán or Jalisco.

Line 320 -322 What is mentioned is relevant. Since production is new in the municipality and wild avocado is used, are these studies relevant to this type of avocado?

Line 335 And to national yield?

Line 345-349 Is it the same setting as your studio?

Line 356-357 Go deeper into the topic

Line 366 I don't think it's nationwide.

Line 387-389 national or foreign consumers?

Line 398-399 It is a confusing conclusion. Is the scenario presented in this study representative of the situation in Mexico? It is important to put similar studies done in the most productive avocado municipalities

Line 399 – 400 They are not developing countries, they are emerging countries.

Line 410-412 This part is worth highlighting in the manuscript, it presents 2 or more scenarios in the case of avocado production and marketing in Mexico. Which one is this?

Line 431-433 This happens in many crops in Mexico.

Conclusions: go deeper into the conclusions

The study is important; however, it presents a scenario that does not correspond to the Mexican avocado. It should be particularized only in the situation of the State of Edomex and in the municipality. What happens with production, marketing, export, etc., is more complicated than the scenario presented in this manuscript.

Reviewer #2: 33: the author referred to export information in 2020 during Covid-19. Such a year was the exception, so I suggest using export data for the time after that. Also, the data was collected during/right after the pandemic, so it is important to discuss the applicability of the results after that.

The introduction needs to expand more, detailing related policies within Mexico and providing information about international trade, such as the trade balance for avocados, and information about the importance of avocados, such as the ratio of planted areas or agricultural production or the contribution to the agricultural GDP. Also, it needs to detail relevant previous studies, identify the gap in the literature, and the contribution of this paper in filling this gap.

78: equation 1 is not clear; please explain more.

79: what does “dose per lot” mean? Not clear. Also, a typo-delete one parenthesis.

132: I suggest starting the sentence with the “private profitability” phrase.

137: it says that “(J=B-F); transfers due to the price of internal factors.” This is inconsistent with table 1 indicating “(J=B-F) as transfer by commercial inputs, not internal factors. Please clarify.

236: not clear how APU2 is more profitable than APU1, while the average harvest is 44kg/ tree, which is lower than that of APU1. Please clarify.

330: The studies by [7, 23, 25] reported…

383: typo within, not withing.

387: it could also be that the avocado demand is inelastic, in which the tax is passed on to the consumers.

Finally, I suggest discussing limitations, policy implications, and future research.

We look forward to receiving your revised manuscript.

Kind regards,

Noé Aguilar-Rivera

Academic Editor

PLOS ONE

Journal Requirements:

2. Please amend either the title on the online submission form (via Edit Submission) or the title in the manuscript so that they are identical.

Reviewers' comments:

Reviewer's Responses to Questions

**Comments to the Author**

1. Is the manuscript technically sound, and do the data support the conclusions?

Reviewer #1: Partly

Reviewer #2: Yes

2. Has the statistical analysis been performed appropriately and rigorously?

Reviewer #1: Yes

Reviewer #2: Yes

3. Have the authors made all data underlying the findings in their manuscript fully available?

Reviewer #1: Yes

Reviewer #2: Yes

4. Is the manuscript presented in an intelligible fashion and written in standard English?

Reviewer #1: Yes

Reviewer #2: Yes

Reviewer #1: Abstract: ok

Introduction: Line 33 Don't you have a more up-to-date figure?

The topic should be delved into if there are studies related to the cultivation of avocado, at the Mexico and state level, or related studies to see the methodological viability in important crops, in Michoacán or Jalisco.

Material and methods: Line 52 – 54 add a location map. Delve into the place of study, if it is important in the production of avocado, what ranking it occupies in the state, among other characteristics that describe the relevance of the crop in the place.

Line 64 What was the density?

Line 115 – 116 Is the product produced in this area imported to the USA? How much?

It is understood that the avocado produced at this site is the same as that produced in Michoacán. The characteristics of avocados in both areas must be different. These differences must be clarified.

On the political side, it is homogenized in the country.

Results: line 201 – 216 this part should go in methodology.

Line 218-220 compare it with the production of Michoacán or Jalisco.

Line 320 -322 What is mentioned is relevant. Since production is new in the municipality and wild avocado is used, are these studies relevant to this type of avocado?

Line 335 And to national yield?

Line 345-349 Is it the same setting as your studio?

Line 356-357 Go deeper into the topic

Line 366 I don't think it's nationwide.

Line 387-389 national or foreign consumers?

Line 398-399 It is a confusing conclusion. Is the scenario presented in this study representative of the situation in Mexico? It is important to put similar studies done in the most productive avocado municipalities

Line 399 – 400 They are not developing countries, they are emerging countries.

Line 410-412 This part is worth highlighting in the manuscript, it presents 2 or more scenarios in the case of avocado production and marketing in Mexico. Which one is this?

Line 431-433 This happens in many crops in Mexico.

Conclusions: go deeper into the conclusions

The study is important; however, it presents a scenario that does not correspond to the Mexican avocado. It should be particularized only in the situation of the State of Edomex and in the municipality. What happens with production, marketing, export, etc., is more complicated than the scenario presented in this manuscript.

Reviewer #2: 33: the author referred to export information in 2020 during Covid-19. Such a year was the exception, so I suggest using export data for the time after that. Also, the data was collected during/right after the pandemic, so it is important to discuss the applicability of the results after that.

The introduction needs to expand more, detailing related policies within Mexico and providing information about international trade, such as the trade balance for avocados, and information about the importance of avocados, such as the ratio of planted areas or agricultural production or the contribution to the agricultural GDP. Also, it needs to detail relevant previous studies, identify the gap in the literature, and the contribution of this paper in filling this gap.

78: equation 1 is not clear; please explain more.

79: what does “dose per lot” mean? Not clear. Also, a typo-delete one parenthesis.

132: I suggest starting the sentence with the “private profitability” phrase.

137: it says that “(J=B-F); transfers due to the price of internal factors.” This is inconsistent with table 1 indicating “(J=B-F) as transfer by commercial inputs, not internal factors. Please clarify.

236: not clear how APU2 is more profitable than APU1, while the average harvest is 44kg/ tree, which is lower than that of APU1. Please clarify.

330: The studies by [7, 23, 25] reported…

383: typo within, not withing.

387: it could also be that the avocado demand is inelastic, in which the tax is passed on to the consumers.

Finally, I suggest discussing limitations, policy implications, and future research.

**Do you want your identity to be public for this peer review?** For information about this choice, including consent withdrawal, please see our Privacy Policy

Reviewer #1: **Yes: ** Olvera-Vargas, Luis Alberto

Reviewer #2: No

---

## [Author Response · Author response to Decision Letter 1]

16 Feb 2025

Reviewer 1:

In response to the reviewers' feedback, the following revisions were made to the original manuscript:

New paragraphs were included:

Lines 32-37 in the revised manuscript: “Mexico leads… …3, 4]”

Lines 38-45: “The avocado… … [6]”.

to reflect updated data.

New paragraphs were added to the introduction to address avocado studies in Mexico and demonstrate the methodology's relevance to key crops at both national and global levels.

Lines 46-52: “In recent… …[8].”

Lines 53-57: “In Mexico,… …[9].”

Lines 58-63: “The State of Mexico… …[3].”

Lines 64-74: “Comparative… ….resource allocation.”

Lines 75-77: “The policy… …[16, 17].”

Lines 78-84: “This methodology… …Malaysia.”

The final paragraph of the introduction was refined for improved clarity.

Lines 85-91: “This study… …region of Mexico.”

A properly cited map showing Ocuilan's relief and coordinates was included.

Line 97: “Figure 1: Map … …. Mexico.”

Ocuilan's temperature range and precipitation were included, along with its ranking in avocado production within the State of Mexico..

Line 95-96: “The area… … agroecological zone.”

Line 109: “120-366 trees per hectare”

It was clarified in the revised manuscript that exports to the U.S. do not occur

Lines 181-183: “Both analyzed… …[9].”

New lines were added to specify that the analyzed avocado systems are small-scale. It is also noted that Mexico's support policies apply uniformly to all avocado systems.

Lines 108: “All APUs were small… …technology.”

Lines 109-111: “The support… …climatic región.”

The paragraphs were moved to the methodology section, under Modelling of production systems. Lines 124-141.

Literature was added to align with studies from Michoacán, Jalisco, and the State of Mexico.

Lines 268-269: “These tree… …36].”

The study focused on commercial plantations (non-subsistence) cultivating Hass avocados. Older plantations are 8–14 years old, while younger ones are 6 years old. Additional details were added to discuss the impact of the Hass variety on native avocados.

Lines 370-372: “A clear trend… …[39].”

The national yield was included.

Line 386: “, while the national… …hectare.”

It is a comparable study conducted in the State of Mexico, but in a different region.

Additional lines were added to delve deeper into the topic.

Lines 410-414: “In 2020,… …transfers.”

This point was clarified in the manuscript:

Lines 109-111: “The support… …climatic región.”

It was clarified that taxes are borne by the domestic consumer.

Line 446: “the end national consumer…”

To avoid confusión the following lines were included:

Lines 455-460: “Small-scale… …[17, 46].”

Developing was changed to emerging.

The scenarios cover both production and marketing.

A paragraph was added to clarify this point.

Lines 484-490: “An effect… …deprivation”.

This indeed occurs with many crops in Mexico.

The conclusion of the study was expanded.

Lines 513-536: “To assess… …this study.”

The study represents 2% of the avocado cultivation area in the State of Mexico but is considered representative of its production system. Consensus panel results reflect not only this 2% but also 44% of the state’s total avocado production.

Reviewer 2:

New paragraphs were added to include post-COVID literature, along with a discussion on the results' applicability.

Lines 32-84: “Mexico leads… …Malaysia.”

Lines 560-562: “The authors… …consumption.”

The introduction was revised to expand the literature and include updated figures. The final paragraph was rewritten to emphasize the significance of this study

Lines 32-84.

Lines 85-91: “This study… …of Mexico.”

The equation was rewritten.

Line 142

“dose per lot” was changed to “dose per surface”.

Line 143

The observation has been addressed.

Line 197

There was indeed an error, which has been corrected in the text.

Line 228 was removed: “Transfers due to the price of internal factors”

The error was corrected: the average is 144 kg, not 44.

Line 267.

The observation has been addressed.

Line 380.

The observation has been addressed.

Our study focuses on evaluating the impact of policy or tax cascades on avocado producers, not elasticity.

To clarify the conclusions, the hypothesis is addressed through two questions, each answered and supported with arguments. Proposals for future research are also included, considering the paper's limitations.

---

## [Decision Letter · Decision Letter 1]

28 Apr 2025

Economic Analysis of Avocado Production Systems: Market Failures and Policy Distortions

PLOS ONE

Dear Dr. Hernandez,

Thank you for submitting your manuscript to PLOS ONE. After careful consideration, we feel that it has merit but does not fully meet PLOS ONE’s publication criteria as it currently stands. Therefore, we invite you to submit a revised version of the manuscript that addresses the points raised during the review process.

We look forward to receiving your revised manuscript.

Kind regards,

Noé Aguilar-Rivera

Academic Editor

PLOS ONE

Reviewers' comments:

Reviewer's Responses to Questions

**Comments to the Author**

Reviewer #1: All comments have been addressed

Reviewer #3: (No Response)

2. Is the manuscript technically sound, and do the data support the conclusions?

Reviewer #1: Yes

Reviewer #3: Yes

3. Has the statistical analysis been performed appropriately and rigorously?

Reviewer #1: Yes

Reviewer #3: No

4. Have the authors made all data underlying the findings in their manuscript fully available?

Reviewer #1: Yes

Reviewer #3: No

5. Is the manuscript presented in an intelligible fashion and written in standard English?

Reviewer #1: Yes

Reviewer #3: Yes

Reviewer #1: Abstract: ok

Introduction: Line 33 Mexico is not a price setter. Line 49 – 50 the problems are stronger in states that want to excel in avocado exports. Line 53 – 54 phytosanitary programs are concentrated in the state that produces and exports the most to the USA.

Materials and methods: Lines 93 – 94 topographical conditions are not comparable to the national average; the most favorable conditions for avocado cultivation are in an altitude range between 1000 and 2300 meters above sea level. Won't this variability of conditions generate an error in your method? In Figure 1, the level curves confuse with the flat areas, it seems that the flattest areas are to the northwest of the municipality. Remove the level curves or the physiographic legend.

Modeling of production systems: the conditions of your UPA are different from those of avocado crops in Michoacán and Jalisco. Consider this information if you are going to make national comparisons. In general, the methodology seems very solid

Results: line 268 – 269 Are the yield values average across the three states? There may be a lot of variability between production types in these three states. The results presented are precise and concrete.

Discussion: lines 378 – 370 This data is relevant to your results. Can it be said that production in this municipality is not economically and socially profitable? When comparing data from other states or countries, it seems that the result is always different. Is this because the crop is relatively new in the area? Line 408 – 409 It is clear that protection for export producers is always more important, including plant health protection. Line 442 -444 This data is relevant. It should be discussed further. Line 469 Could it be that avocado cultivation is not favorable for small farmers? Your discussion highlights that international and export trade is not viable or favorable for poor farmers, or at least Mexico's public policies do not favor them.

Conclusions: clear and consistent with the results and discussion

Reviewer #3: This manuscript presents a notable strength by collecting primary data directly from 11 farmers and using numerical evidence to evaluate the impact of agricultural policies. The application of the Policy Analysis Matrix (PAM) effectively highlights distortions between private and social prices. Additionally, the comparison of Mexican agricultural subsidies with those in the United States and Canada provides insightful discussion on policy inefficiencies.

The study is timely and relevant, but I would like to suggest the following points for revision and clarification:

1. Please consider including the standard deviations for the data obtained from the 11 surveyed farms. This would enhance the reliability of the analysis. This could be presented in the Supplemental Data if space is limited in the main text.

1. The resolution of the map provided is low, making the legend difficult to read. A higher-resolution version would be appreciated.

2. The term "CNMPP" is used without a clear definition in the main text. Please provide an explanation.

3. While "NPCI" is defined in the Materials and Methods section, it does not appear to be reported in the Results. Please clarify or include the relevant results.

4. Results – Yield (Line 265–): The fruit quality classification is said to follow reference [37], but the explanation is not sufficiently clear. Please provide more detail on how the classification was carried out.

5. Tables 2–4: The captions should include a brief explanation of the term “Divergences” for better reader understanding.

6. Tables 3–4: Some formatting issues are present in the tables. Please revise to ensure all contents are clearly readable.

7. Tables 2–4: Presenting the information for APU1 to APU3 in a single consolidated table may improve readability. Alternatively, showing the profitability by tree age using a time-series graph could help better illustrate the policy impact.

8. Table 5: Please revise the column header “Age” to “Age of Avocado Trees” for greater clarity.

9. Line 309: The reference to “Table 1, 2, and 3” should be corrected to “Table 2, 3, and 4.”

10. Line 309: The sentence “[The main outlays, constituting 43%, 48%, 55% of total costs respectively, were for fertilizers.]” is unclear. Which figure or table supports this claim? If the data are not presented, please consider including them.

11. Discussion: The paper attributes market failure primarily to low wages. However, could other factors also be contributing? A more nuanced discussion would strengthen the argument.

12. Discussion: The authors mention that support for smallholder farmers is necessary. It would be helpful to suggest specific policy measures or practical approaches to make this recommendation more actionable.

Overall, this is a valuable contribution, and I believe the above revisions will significantly improve the clarity, rigor, and practical relevance of the manuscript.

**Do you want your identity to be public for this peer review?** For information about this choice, including consent withdrawal, please see our Privacy Policy

Reviewer #1: **Yes: ** Luis Alberto Olvera-Vargas

Reviewer #3: No

---

## [Author Response · Author response to Decision Letter 2]

9 Jun 2025

In response to the reviewers' feedback, the following changes were made to the original manuscript:

Reviewer 1:

Introduction: Line 33 Mexico is not a price setter. Line 49 – 50 the problems are stronger in states that want to excel in avocado exports

Response:

Line 33 was removed, and a new paragraph was added to highlight the related issue.

Lines 48-54: “Avocado cultivation… … [4, 6].”

Reviewer:

Line 53 – 54 phytosanitary programs are concentrated in the state that produces and exports the most to the USA.

Response:

The suggested line was included in the manuscript.

Lines 60-61: “Phytosanitary… … USA.”

Reviewer: Materials and methods: Lines 93 – 94 topographical conditions are not comparable to the national average; the most favorable conditions for avocado cultivation are in an altitude range between 1000 and 2300 meters above sea level. Won't this variability of conditions generate an error in your method?

Response: The altitude value for the study region has been corrected, including only the altitude range of producing regions.

Lines 97-99: “Between… …sea level.”

Reviewer:In Figure 1, the level curves confuse with the flat areas, it seems that the flattest areas are to the northwest of the municipality. Remove the level curves or the physiographic legend.

Response:Figure 1 was modified to avoid confusion.

Reviewer: Modeling of production systems: the conditions of your UPA are different from those of avocado crops in Michoacán and Jalisco. Consider this information if you are going to make national comparisons. In general, the methodology seems very solid

Response: It was clarified that the system operates under rain-fed conditions to avoid confusion with other methods used nationwide.

Lines 137-138: “These… … rainfed.”

Reviewer:Results: line 268 – 269 Are the yield values average across the three states? There may be a lot of variability between production types in these three states. The results presented are precise and concrete.

Response: The paragraph was revised to emphasize the differences in average avocado yield according to tree age.

Lines 273-277: “Plantations… … [20, 36].”

Reviewer:Discussion: lines 378 – 370 This data is relevant to your results. Can it be said that production in this municipality is not economically and socially profitable? When comparing data from other states or countries, it seems that the result is always different. Is this because the crop is relatively new in the area?

Response: A new sentence was added to further elaborate on the profitability of avocado production.

Lines 392-394: “Developing… … profitable.”

Reviewer:Line 408 – 409 It is clear that protection for export producers is always more important, including plant health protection.

Response:A sentence was added to address the comment.

Lines 456-457: “This… … [16].”

Reviewer:Line 442 -444 This data is relevant. It should be discussed further.

Response: The idea was expanded to emphasize the impact of indirect taxes on small-scale avocado producers.

Lines 463-469: “These indirect… … mechanisms.”

Reviewer:Line 469 Could it be that avocado cultivation is not favorable for small farmers? Your discussion highlights that international and export trade is not viable or favorable for poor farmers, or at least Mexico's public policies do not favor them.

Response:A sentence was included to highlight the importance of small-scale avocado farmers.

Line 494: “However… … fruit.”

Reviewer 3

Reviewer:Please consider including the standard deviations for the data obtained from the 11 surveyed farms. This would enhance the reliability of the analysis. This could be presented in the Supplemental Data if space is limited in the main text.

Response:A new table was included with weighted standard deviation based on technical coefficients.

Line 280: “Table 2…. …. coefficients.”

Reviewer:The resolution of the map provided is low, making the legend difficult to read. A higher-resolution version would be appreciated.

Response:Figure 1 was modified for better understanding of the analyzed region. The Legend was simplified, and text resolution was improved.

Line 103

Reviewer:The term "CNMPP" is used without a clear definition in the main text. Please provide an explanation.

Response:The correct term is NPCI (Nominal Protection Coefficient for Marketable Inputs), and this abbreviation has been applied consistently across the manuscript.

Reviewer:While "NPCI" is defined in the Materials and Methods section, it does not appear to be reported in the Results. Please clarify or include the relevant results.

Response: There was some confusion regarding the term CNMPP; it has been replaced with NPCI, and the corresponding paragraph has been added.

Line 347: “Nominal… … (NPCI).

Reviewer:Results – Yield (Line 265–): The fruit quality classification is said to follow reference [37], but the explanation is not sufficiently clear. Please provide more detail on how the classification was carried out.

Response:The classification in accordance with Mexican regulations was expanded, and a table was added.

Line 291: “Table 3. Fruit…

Reviewer: Tables 2–4: The captions should include a brief explanation of the term “Divergences” for better reader understanding

Response: The tables including the term “divergences” were removed and a figure was included as replacement.

A brief explanation of the term “divergence” is included in lines 180-182.

Reviewer:Tables 3–4: Some formatting issues are present in the tables. Please revise to ensure all contents are clearly readable.

Response: The tables mentioned were removed.

Reviewer: Tables 2–4: Presenting the information for APU1 to APU3 in a single consolidated table may improve readability. Alternatively, showing the profitability by tree age using a time-series graph could help better illustrate the policy impact.

Response: Tables 2, 3, and 4 were removed, and a new figure was added to summarize the results.

Line 302: “Figure 2. Profitability…”.

Reviewer: Table 5: Please revise the column header “Age” to “Age of Avocado Trees” for greater clarity.

Response:The comment has been addressed.

Reviewer:Line 309: The reference to “Table 1, 2, and 3” should be corrected to “Table 2, 3, and 4.”

Response: The tables mentioned were removed.

Reviewer: Line 309: The sentence “[The main outlays, constituting 43%, 48%, 55% of total costs respectively, were for fertilizers.]” is unclear. Which figure or table supports this claim? If the data are not presented, please consider including them.

Response: A figure was added to support the argument.

Lines 322: “Figure 3. Marketable…”

Reviewer: Discussion: The paper attributes market failure primarily to low wages. However, could other factors also be contributing? A more nuanced discussion would strengthen the argument.

Response: The discussion was expanded to include additional factors that may contribute to avocado market failures.

Lines 534-542: “Market failures… … [54].”

Reviewer: The authors mention that support for smallholder farmers is necessary. It would be helpful to suggest specific policy measures or practical approaches to make this recommendation more actionable.

Response: A practical focus was incorporated into the recommendations.

Lines 516-519: “Addressing… …. Development.”

---

## [Decision Letter · Decision Letter 2]

2 Jul 2025

Economic Analysis of Avocado Production Systems: Market Failures and Policy Distortions

PLOS ONE

Dear Dr. Hernandez,

Thank you for submitting your manuscript to PLOS ONE. After careful consideration, we feel that it has merit but does not fully meet PLOS ONE’s publication criteria as it currently stands. Therefore, we invite you to submit a revised version of the manuscript that addresses the points raised during the review process.

https://journals.plos.org/plosone/s/submission-guidelines#loc-laboratory-protocols . Additionally, PLOS ONE offers an option for publishing peer-reviewed Lab Protocol articles, which describe protocols hosted on protocols.io. Read more information on sharing protocols at https://plos.org/protocols?utm_medium=editorial-email&utm_source=authorletters&utm_campaign=protocols .

We look forward to receiving your revised manuscript.

Kind regards,

Noé Aguilar-Rivera

Academic Editor

PLOS ONE

Journal Requirements:

Reviewers' comments:

Reviewer's Responses to Questions

**Comments to the Author**

Reviewer #1: All comments have been addressed

Reviewer #4: (No Response)

2. Is the manuscript technically sound, and do the data support the conclusions?

Reviewer #1: Yes

Reviewer #4: Yes

3. Has the statistical analysis been performed appropriately and rigorously?

Reviewer #1: Yes

Reviewer #4: N/A

4. Have the authors made all data underlying the findings in their manuscript fully available?

Reviewer #1: Yes

Reviewer #4: Yes

5. Is the manuscript presented in an intelligible fashion and written in standard English?

Reviewer #1: Yes

Reviewer #4: Yes

Reviewer #1: The authors made improvements and comments. They justified questions and improved the discussion for a better understanding of the manuscript.

Reviewer #4: (No Response)

**Do you want your identity to be public for this peer review?** For information about this choice, including consent withdrawal, please see our Privacy Policy

Reviewer #1: **Yes: ** Luis Alberto Olvera-Vargas

Reviewer #4: No

---

## [Author Response · Author response to Decision Letter 3]

25 Jul 2025

In response to the reviewers' feedback, the following changes were made to the original manuscript:

Reviewer: Line 131, “with ranges” should be replaced with “with age ranges”.

Answer: The correction has been addressed.

R:Line 280, you mentioned in Table 2 for APU2 (7-9), APU3 (10 to 14), in Table 3 for APU3 (7m10a14) and in Table 5 for APU2 (6-9), APU3 (>10), with respective age ranges, but they are not in line with what in materials and methods (Line 132-133, where APU1 (4-6 years), APU2 (8-9 years), APU3 (10-15 years). Their descriptions should be consistent throughout the paper

A: The terminology was homogenized throughout the manuscript.

R: In Table 2 (Line 280), there is a misspelling, “tres”, which should be “trees”.

A: The correction has been addressed.

R:The names of Table 1, 2, and 3 are too short and not self-explained, which need a bit more words.

A: The title of tables 1, 2 and 3 were modified.

R:Figure 2 does not show any unit.

A: Units were included in figure 2.

R:Line 498, 542, the quotes (“”) are not necessary for the words “harms” and “distorting”

A. The correction has been addressed.

R.You referred Table 6 in Lines, 341, 354, 459, 467, 375, but I could not find Table 6 anywhere.

A:The table that was referred was Table 5 not 6

R:In Table 4, you need the unit (m or meter) in column 2.

A:The unit of measure for Table 4 was included.

R:In Figure 3, the APUs is described in title, but it was not included in figure, instead you mentioned 6m4a6, 6m7a9 and 7m10a14, for which I would suggest just to write APU1, APU2 and APU3 simply.

A:The legend in figure 3 was modified.

---

## [Editor Report · Decision Letter 3]

31 Jul 2025

Economic Analysis of Avocado Production Systems: Market Failures and Policy Distortions

PONE-D-24-45504R3

Dear Dr. Elein Hernandez

We’re pleased to inform you that your manuscript has been judged scientifically suitable for publication and will be formally accepted for publication once it meets all outstanding technical requirements.

Kind regards,

Noé Aguilar-Rivera

Academic Editor

PLOS ONE
---

## [Editor Report · Acceptance letter]

PONE-D-24-45504R3

PLOS ONE

Dear Dr. Hernandez,

I'm pleased to inform you that your manuscript has been deemed suitable for publication in PLOS ONE. Congratulations! Your manuscript is now being handed over to our production team.

Kind regards,

on behalf of

Dr. Noé Aguilar-Rivera

Academic Editor

PLOS ONE